# Determination of Aminoglycosides by Ion-Pair Liquid Chromatography with UV Detection: Application to Pharmaceutical Formulations and Human Serum Samples

**DOI:** 10.3390/molecules29133210

**Published:** 2024-07-05

**Authors:** Eliseo Herrero-Hernández, Diego García-Gómez, Irene Ramírez Pérez, Encarnación Rodríguez-Gonzalo, José Luis Pérez Pavón

**Affiliations:** Department of Analytical Chemistry, Nutrition and Food Science, University of Salamanca, Plaza de los Caídos s/n, 37008 Salamanca, Spain; elihh@usal.es (E.H.-H.); dgg@usal.es (D.G.-G.); irene_rp@usal.es (I.R.P.); erg@usal.es (E.R.-G.)

**Keywords:** aminoglycosides, antibiotics, ion-pair liquid chromatography, pharmaceutical formulations, human serum

## Abstract

Aminoglycosides (AGs) represent a prominent class of antibiotics widely employed for the treatment of various bacterial infections. Their widespread use has led to the emergence of antibiotic-resistant strains of bacteria, highlighting the need for analytical methods that allow the simple and reliable determination of these drugs in pharmaceutical formulations and biological samples. In this study, a simple, robust and easy-to-use analytical method for the simultaneous determination of five common aminoglycosides was developed with the aim to be widely applicable in routine laboratories. With this purpose, different approaches based on liquid chromatography with direct UV spectrophotometric detection methods were investigated: on the one hand, the use of stationary phases based on hydrophilic interactions (HILIC); on the other hand, the use of reversed-phases in the presence of an ion-pairing reagent (IP-LC). The results obtained by HILIC did not allow for an effective separation of aminoglycosides suitable for subsequent spectrophotometric UV detection. However, the use of IP-LC with a C18 stationary phase and a mobile phase based on tetraborate buffer at pH 9.0 in the presence of octanesulfonate, as an ion-pair reagent, provided adequate separation for all five aminoglycosides while facilitating the use of UV spectrophotometric detection. The method thus developed, IP-LC-UV, was optimized and applied to the quality control of pharmaceutical formulations with two or more aminoglycosides. Furthermore, it is demonstrated here that this methodology is also suitable for more complex matrices, such as serum, which expands its field of application to therapeutic drug monitoring, which is crucial for aminoglycosides, with a therapeutic index ca. 50%.

## 1. Introduction

Aminoglycosides (AGs) are a group of water-soluble, polar and cationic antibiotics characterized by their unique chemical structures and potent bactericidal activity. They were introduced in 1943 when streptomycin was isolated from *Streptomyces griseus* [1]. Aminoglycosides show a broad spectrum, uniform pharmacokinetic behaviour and low cost. Therefore, their clinical significance in the treatment of life-threatening infections has been fundamental [2,3,4]. However, their use is not without drawbacks. Because of the relatively narrow therapeutic window for AGs, they suffer from the frequent occurrence of ototoxicity and nephrotoxicity, which have decreased their use. In this sense, their efficiency is closely linked to achieving and maintaining therapeutic drug concentrations while avoiding toxic levels [5]. Subsequently, the precise monitoring of AG concentrations in patient serum, thereby optimizing dosing regimens and minimizing the risk of adverse events, is mandatory. Furthermore, the emergence of antibiotic-resistant strains of bacteria, including those resistant to AGs, highlights the need for precise and reliable analytical methods for monitoring and ensuring the appropriate use of these antibiotics. Streptomycin is on the WHO’s list of essential medicines and AGs are still in use for the treatment of human and veterinary bacterial infections [6].

The analysis of AGs faces certain difficulties because of their high polarity, their polycationic character and the lack of chromophores, which results in very poor behaviour in reversed-phase LC with direct UV detection. Not only do AGs show poor retention, but they also suffer from poor responses in the UV detector, which leads to detection limits, usually above 100 mg L^−1^. Chromatographic and spectrophotometric methodologies are cost-effective and streamlined procedures, representing the gold-standard technique for the determination of drugs in the pharmaceutical industry. The spectrophotometric approach is favoured for its straightforward protocols, widespread instrument availability, precision and accuracy. Nevertheless, challenges in the experimental evaluation of spectrophotometry arise from the necessity of larger sample quantities for analysis and constraints in detecting elevated sample concentrations. Furthermore, complications arise when addressing the overlapping spectral bands of analytes and interferences, leading to complexities in quantitative data analysis [7,8].

Several alternatives to the standard approach based on reversed-phase LC have been proposed to overcome the above-mentioned difficulties, such as the use of hydrophilic interaction liquid chromatography (HILIC) [9] and ion-pair liquid chromatography (IP-LC). An ad hoc column has even been developed for the analysis of streptomycin [10]. Several interesting reviews on the challenges in the determination of these antibiotics by HPLC were published by Farouk et al. [11] and Zang et al. [12].

As mentioned, several papers have reported protocols employing hydrophilic interaction liquid chromatography (HILIC) for the separation of AGs with stationary phases such as bare silica, diol, etc. [13,14,15,16,17,18]. Better results seem to be obtained by zwitterionic stationary phases, based on the presence of zwitterionic groups (sulfobetaine, phosphorylcholine) covalently bound to the surface of silica particles, which combine retention by hydrophilic partitioning and by weak ionic interactions [19,20,21,22,23]. Regarding detection, as HILIC allows the separation of AGs in their native forms, subsequent detection by UV, due to the lack of chromophores in AGs, is unfeasible.

Ion-pair liquid chromatography (IP-LC) is based on the addition of an appropriate ion-pairing agent to the mobile phase to enhance hydrophobicity and promote retention in reversed-type stationary phases. The positively charged amino groups present in the AG antibiotics interact with negatively charged counterions in the mobile phase, leading to the formation of hydrophobic ion pairs. Common agents used for AG determination include alkyl sulfonate and carboxylate ions [24,25,26,27]. However, the presence of ion-pairing agents in IP-LC suppresses the ionization process when ESI-MS detection is used. Additionally, MS detection greatly increases the cost of analysis and is only compatible with volatile additives. As an alternative, IP-LC can be coupled with other detection techniques. IP-LC with spectrophotometric UV detection for AGs has been reported after chemical derivatization pre- or post-column to solve the lack of chromophores [28,29,30,31]. However, some derivatizing agents can react with different groups of AGs producing multiple derivatives from a single analyte. Furthermore, the efficiency of derivatization largely depends on the composition of the sample matrix, especially in the case of complex matrices such as biological fluids. Overall, methods involving derivatization are more commonly applied to analyze samples in simple matrices or when only a single analyte is determined [29,30,31].

A different derivatization process based on the use of borate complexation was described by Flurer [32] for the analysis of AGs by capillary electrophoresis with UV detection. The aminoglycoside–borate complex altered electrophoretic migration times and also allowed direct UV detection of these compounds. This strategy was adapted by Blanchaert et al. [33] for the assay of Kanamycin A by HPLC-UV using a mobile phase containing octanesulfonate in addition to borate. The same authors reported its successful application to the analysis of Tobramycin and Amikacin, but unsatisfactory results were obtained in the case of Gentamicin [34]. This approach was applied to the analysis of pharmaceutical formulations containing a single analyte as active ingredient.

In this work, we further explore the addition of borate ions to the mobile phase to facilitate the use of UV detection coupled with IP-LC for the determination of aminoglycoside antibiotics of common use. Streptomycin, Spectinomycin, Neomycin, Kanamycin and Dihydrostreptomycin were selected as target analytes.

The final objective is the development of an analytical method based on LC-UV for the joint determination of several AGs of wide applicability, i.e., that can be easily used in routine laboratories both for quality control in pharmaceutical formulations and for the therapeutic monitoring of these drugs in biological samples. In this sense, UV detectors were the choice, since they are simple and easy to use and, therefore, LC-UV equipment is commonly present in most pharmaceutical laboratories.

## 2. Results and Discussion

As mentioned above, the high polarity and polycationic character of AGs makes their retention in conventional reversed-phase stationary phases a difficult task. Here, we investigate HILIC and IP-LC for adequate separation of these compounds, always keeping in mind our initial intention to subsequently apply UV detection.

### 2.1. Study of the Separation of Aminoglycoside Antibiotics by HILIC

HILIC is regarded as the appropriate chromatographic mode when the retention of hydrophilic compounds, such as aminoglycosides, is required.

However, retention behaviour in HILIC is difficult to predict, since different types of interactions are involved (e.g., hydrophilic partitioning, polar interactions and ionic interactions). This means that the selectivity obtained is affected by multiple factors, including the nature of the stationary phase and the composition of the mobile phase, including the percentage and nature of the organic component, the pH, the concentration and ionic strength of the buffer, and the temperature.

Different experiments were carried out to find the stationary and mobile phases that would allow for good separation for the analytes selected in the study. Two HILIC stationary phases were tested, one with and the other without permanent charges: the first was an amide type (XBridge^®^ Amide 3.5 µm), and the second was of a zwitterionic nature (SeQuant^®^ ZIC^®^-HILIC, 3.5 µm).

Figure 1 shows the LC-UV chromatograms obtained when individual standards of each analyte were injected in both stationary phases, XBridge^®^ Amide (Figure 1a) and ZIC^®^-HILIC (Figure 1b). Both the amide-type and zwitterionic columns showed insufficient selectivity for AGs, all the compounds eluted very close to each other. This makes the use of UV detection unfeasible to address any mixture. Different elution conditions were tested to improve selectivity, i.e., several mobile phases based on mixtures of acetonitrile and ammonium formate 100 mM or formic acid 0.1%, different initial compositions for the mobile phase and different gradients, including isocratic steps. Nevertheless, no satisfactory results were obtained. Finally, an HILIC/SAX/WCX trimodal phase column (Acclaim Trinity P2 column) was also evaluated, with a mixture of acetonitrile and ammonium formate 100 mM as the mobile phase. Similarly, an effective separation of the five AGs was not achieved, and poor peak morphology was observed.

### 2.2. Study of the Separation of Aminoglycoside Antibiotics by IP-LC

Another approach to separate hydrophilic and ionic compounds is the use of RPLC with ion-pairing agents (IP-LC).

First, the use of heptafluorobutyric acid (HFBA) as an ion-pairing agent was evaluated. An aqueous mobile phase containing 0.1% HFBA combined with a C18 stationary phase and UV detection at 200 nm was tested. In this case, good separation was achieved for two of the antibiotics, SPM and NMC. However, STM and DHS were not resolved. Additionally, the peak corresponding to KNC was not observed, probably due to coelution with the signal corresponding to HFBA.

A second attempt to obtain an adequate chromatographic separation, aimed at developing a method that allows the use of UV detection, consisted of applying the procedure reported by Blanchaert et al. [31,32] based on the formation of complexes with borate and the subsequent formation of an ion pair with octanesulfonate. These authors described the formation of a complex of certain AGs containing vicinal diols with borate anion at pH levels higher than 8. On the one hand, the presence of borate favoured the UV detection of AGs by improving UV absorption. On the other hand, octanesulfonate acts as an ionic-pair-forming agent, improving their retention and separation by reversed-phase LC. Satisfactory results were initially reported for Kanamycin [31] and later for Tobramycin and Amikacin, but inadequate results were found in the case of Gentamicin [32]. To our knowledge, the application of this procedure to other AG antibiotics has not yet been explored.

Figure 2 shows the IP-LC-UV chromatograms corresponding to the injection of a mixture of the five AGs at a concentration of 100 mg mL^−1^. An aqueous mobile phase containing a 25 mM aqueous disodium tetraborate decahydrate buffer (pH = 9.0) and 0.1% (*w*/*v*) sodium octanesulfonate was used (top blue line).

The effect of the borate buffer can be seen in the chromatogram immediately below (red line in Figure 2) that was obtained from a mobile phase not containing that buffer. It should be noted that poor signals were obtained for some of the analytes, such as STR and DHST. For the rest of the analytes, UV absorption was too weak to even detect the analytes.

Finally, from a mobile phase without octanesulfonate (green line in Figure 2), a single unretained peak appeared at the dead volume, corresponding to the elution of the five AGs without separation. The bottom purple line in Figure 2 corresponds to the chromatogram corresponding to a mobile phase containing neither borate nor octanesulfonate.

These results showed that the presence of both reagents (borate and octanesulfonate) in the mobile phase was necessary to obtain good retention and sensitivity by IP-LC-UV.

### 2.3. Validation of the Optimized IP-LC-UV Procedure

The developed IP-LC-UV methodology was validated for each of the AGs by studying the linear range, LODs and LOQs, accuracy (average recovery) and precision (reproducibility and repeatability). All compounds displayed good linearity over the selected range (10–1000 mg L^−1^) with regression coefficients higher than 0.99 (analytical characteristics of the method are shown in Table 1). The LODs and LOQs were estimated as the analyte concentration corresponding to a signal-to-noise ratio of 3 and 10, respectively. LODs ranged between 3 and 64 mg L^−1^ and LOQs ranged between 14 and 192 mg L^−1^ for DHS and NMC, respectively (Table 1).

The accuracy of the method was evaluated by analyzing spiked water samples at three levels and quantifying them with the previous calibration curve by external standard calibration. Relative errors were less than 8.3% for all compounds and concentrations. The intra- and the inter-day precision for AGs were calculated at two concentration levels, the respective LOQs and 1000 mg L^−1^. The method showed good intra- and inter-day precision at both concentration levels, with values below 9.9% and 3.5% for repeatability and 17.2% and 6.8% for reproducibility. After comparison with the background noise in various matrices, the results demonstrated that there were no interfering peaks at the expected retention times for the target analytes, which confirms the specificity of the method.

### 2.4. Analysis of Pharmaceutical Formulations

The developed IP-LC-UV method can be successfully applied for the quality control of pharmaceutical formulations containing AGs. As proof of its applicability, we carried out the analysis of two commercially available formulations: The first formulation assayed was a generic bactericidal antibiotic for intramuscular administration used to treat tuberculosis, streptococcal endocarditis, urinary infections and gonorrhea; orally, it is also effective against diarrhea and enteritis, and its formulation was a powder for injection containing 1 g of streptomycin sulphate. The second formulation assayed was Sulfintestin Neomycin^®^, indicated for the treatment of enteritis, acute gastroenteritis, enterocolitis and summer diarrhea, presented in the form of tablets, and containing 21 mg of Dihydrostreptomycin sulphate and 39 mg of Neomycin sulphate per tablet.

In both cases, the active ingredients were extracted in water under agitation and the aqueous extracts were analyzed with the developed IP-LC-UV method. Figure 3 shows the chromatograms obtained for both formulations. No interferences caused by the excipients were observed, although extra peaks due to matrix components were present. In any case, a perfect baseline separation was obtained for both formulations.

For the injection powder, the concentration of API was determined to be 98.2% of the reported dose. The accuracy for this application was further evaluated by spiking the extracts with STR to final concentrations 110% and 125% of the reported dose. Each sample was analyzed twice. Recoveries were 100.3% and 100.8% respectively, with RSDs of 1% (Table 2). For Sulfintestin Neomycine^®^ tablets, a tablet was weighted, ground and dissolved in water to a final volume of 50 mL. This procedure was carried out in duplicate. The concentrations of DHS and NMC found in these tablets were 106.4 and 92.7% of the reported doses, respectively. Recoveries found after further spiking the extracts at 110% and 125% of the reported dose were 99.4% and 100.3% for DHS and 98.5 and 99.2% for NMC, respectively. The RSDs for each set were below 1% for DHS and 1.5% for NMC (Table 2).

### 2.5. Application to the Analysis of Serum Samples

In addition to the quality control of formulations, the IP-LC-UV method developed here was also applied for therapeutic drug monitoring, which is critical for AGs with a therapeutic range of ca. 50%. In this way, two of the AGs were selected (those with higher retention times and lower LOD and LOQ values), STM and DHS, to study their determination in serum samples. First, a sample pretreatment step involving protein precipitation was studied. Several reagents were assayed for this purpose, i.e., acetonitrile, different acidic solutions of trichloroacetic acid (TCA, 5% *w*/*v*) or metaphosphoric acid (MPA, 10% *w*/*v*) and an extraction buffer (EB) of pH 4 composed of ammonium acetate, EDTA, sodium chloride and trichloroacetic acid. Figure 4 shows the chromatograms obtained for each precipitation agent.

Similar results were obtained when TCA, MPA or EB were used to precipitate the serum proteins. The recoveries obtained in all cases were similar and over 90%. For this reason, TCA solution was selected as the protein precipitation agent. The proportion of sample/TCA solution was subsequently optimized. Three different proportions were tested (the addition of 0.5, 1.0 and 2.0 mL of TCA solution per mL of serum). The recoveries obtained in all cases were similar. For this reason, a sample/TCA solution ratio of 2:1 was selected to avoid excessive sample dilution.

Figure 5 shows the chromatograms corresponding to a standard of STR and DHS at a concentration of 25 mg L^−1^ in TCA (blue line), and a pretreated serum sample spiked before (red line) and after protein precipitation (green line). As can be seen, the signals corresponding to STR and DHS were similar in all cases, indicating that there are not any losses of the analytes during the pretreatment step. AGs’ response linearity, LODs and LOQs were determined from the injection of six serum samples spiked with both analytes in the 10–500 mg L^−1^ concentration range. Both compounds showed good linearity over the selected range with slopes similar to those obtained in UHQ water, which indicates the absence of a matrix effect.

## 3. Materials and Methods

### 3.1. Chemicals and Reagents

The chemical structures of the aminoglycosides included in this study are shown in Figure 6. Spectinomycin dihydrochloride pentahydrate (SPM) was supplied by Sigma–Aldrich (Buchs, Switzerland). Dihydrostreptomycin sesquisulphate hydrate (DHS), Kanamycin sulphate, (KNC) Neomycin solution 10 mg/mL (NMC) and streptomycin sulphate (STM) were obtained from Sigma (Steinheim, Germany). Stock standard solutions of AGs were prepared in ultra-high-quality (UHQ) water at a level of 5000 mg L^−1^ (Neomycin, at 10,000 mg L^−1^) and stored at −20 °C. Standard working solutions were prepared daily, by appropriate dilutions in UHQ water of the stock solutions.

Acetonitrile (ACN) of HPLC grade was supplied by Merck (Darmstadt, Germany). Disodium tetraborate decahydrate of analytical grade was supplied by Prolabo (Fontenay, France) and sodium octanesulfonate was supplied by ACROS Organics. UHQ water was produced with a MilliQ water system by Merck-Millipore. All other chemicals used (trichloroacetic acid (TCA), metaphosphoric acid (MPA), ethylenediaminetetraacetic acid (EDTA), ammonium formate, formic acid and heptafluobutyric acid (HFBA)) were of analytical reagent grade.

### 3.2. LC System and Chromatographic Conditions

Chromatographic separation was performed on an Agilent 1200 Series High-Performance Liquid Chromatography system (Waldbronn, Germany), equipped with a binary pump, a membrane degasser, an autosampler and a diode array detector (DAD). The column was a Cortecs^®^ C18 4.6 × 50 mm, 2.7 μm from Waters (Milford, MA, USA). Optimized IP-LC conditions were as follows: the mobile phase consisted of a 25 mM aqueous solution of disodium tetraborate decahydrate buffer (pH = 9.0) with 0.1% (*w*/*v*) sodium octanesulfonate (A) and acetonitrile (B). The elution was performed with 0% of B kept for 3 min, increased to 10% in 7 min and to 20% in 2 min, returned to 0 % in 1 min and maintained for 5 min for column equilibration. The flow rate of the mobile phase was 0.3 mL min^−1^ with the column thermostated at 30 °C, and the injected volume was 20 µL. The total run time was 18 min. The diode array detector (DAD) was set at 200 nm.

HILIC chromatographic columns assayed were an XBridge^®^ Amide, based on ethylene bridged hybrid (BEH) particle technology of 3.5 µm (4.6 × 150 mm, Waters, Milford, CT, USA), and a zwitterionic column (SeQuant^®^ ZIC^®^-HILIC, 3.5 µm, 100 Å, 4.6 × 150 mm, Merck, Darmstadt, Germany). An HILIC/SAX/WCX trimodal phase column Acclaim Trinity P2, 3 µm (2.1 × 100 mm, Thermo Scientific, Waltham, MA, USA), was also tested.

### 3.3. Validation of the IP-LC-UV Method

The IP-LC-UV method developed was validated by evaluating the following parameters: the linearity of the calibration curves, the limits of detection and quantification, the repeatability and the reproducibility. Repeatability was evaluated as intra-day precision by analyzing 8 samples and calculating the relative standard deviations (RSDs) of the peak area values obtained for each of the compounds studied. Reproducibility was evaluated as precision on different days (inter-day) by analyzing the samples on 5 consecutive days with 4 injections each day. To estimate linearity, LOD and LOQ, eight standards were prepared in UHQ water spiked with all the analytes in the 10–1000 mg L^−1^ concentration range. The LODs and LOQs were estimated as the analyte concentration with a signal-to-noise ratio of 3 and 10, respectively.

### 3.4. Analysis of Pharmaceutical Formulations

To study the applicability of the developed IP-LC-UV method, two different pharmaceutical formulations were tested: Sulfintestin Neomycine^®^ from Normon Laboratories (Madrid, Spain), in the form of tablets, containing 21 mg of Neomycin (sulphate) and 39 mg of Dihydrostreptomycin (sulphate) per tablet, and a generic formulation presented as powder for injection containing 1 g of streptomycin (sulphate) from Reig Jofré Laboratories (Barcelona, Spain). For Sulfintestin Neomycine^®^, a tablet was weighed and crushed individually in a mortar. For streptomycin (sulphate), a fraction of the injectable powder was taken directly. In both cases, the weighed quantity was dissolved in UHQ water to obtain solutions with concentrations in the upper part of the calibration range. The solutions obtained were filtered through 0.22 µm nylon filters and 1 mL of each solution was placed in HPLC vials for their corresponding analysis in triplicate.

### 3.5. Analysis of Streptomycin (STR) and Dihydrostreptomycin (DHS) in Serum Samples

The developed IP-LC-UV method was also applied for therapeutic drug monitoring in a sample of human serum from human male AB plasma, USA origin, and sterile-filtered (Sigma-Aldrich).

Samples were spiked with STR and DHS and frozen at −20 °C. Prior to analysis, frozen serum samples were thawed at room temperature. To precipitate proteins and extract AGs, a TCA solution (5% *w*/*v*) with a ratio of 2:1 (*v*/*v*) (typically 1 mL of serum) was employed. After the addition of TCA, serum samples were vortexed vigorously for 1–2 min to ensure complete protein denaturation and precipitation. The mixtures were then centrifuged at 2800× g for 10 min to pellet the precipitated proteins. The supernatants containing the AGs were carefully separated from the protein pellets and transferred to clean, labelled liquid chromatography vials prior to their chromatographic analysis.

AG levels in the serum samples were quantified based on peak areas obtained from the chromatographic analysis. Calibration curves generated from serum samples spiked with standards with known concentrations were used.

## 4. Conclusions

A method was developed based on IP-LC-UV for the determination of AGs, using a mobile phase formed by acetonitrile and an aqueous solution of sodium tetraborate 25 mM at pH 9 and 1 g L^−1^ of octanesulfonate. The retention mechanism was investigated confirming that the presence of octanesulfonate favours the retention of AGs in reversed-phase conditions via ion-pair formation, while the presence of tetraborate favours the UV detection of AGs by borate complexation at basic pH levels.

Two main advances were achieved: (i) the method proposed here is multicomponent, allowing the joint determination of five AGs in a single run, and (ii) it was applied to the analysis of these compounds in pharmaceutical formulations and serum samples.

These characteristics make the IP-LC-UV method proposed here an adequate and easy-to-use tool for different purposes, such as the quality control of formulations and for therapeutic drug monitoring, which is essential for AGs with a therapeutic range of ca. 50%.

## Figures and Tables

**Figure 1 molecules-29-03210-f001:**
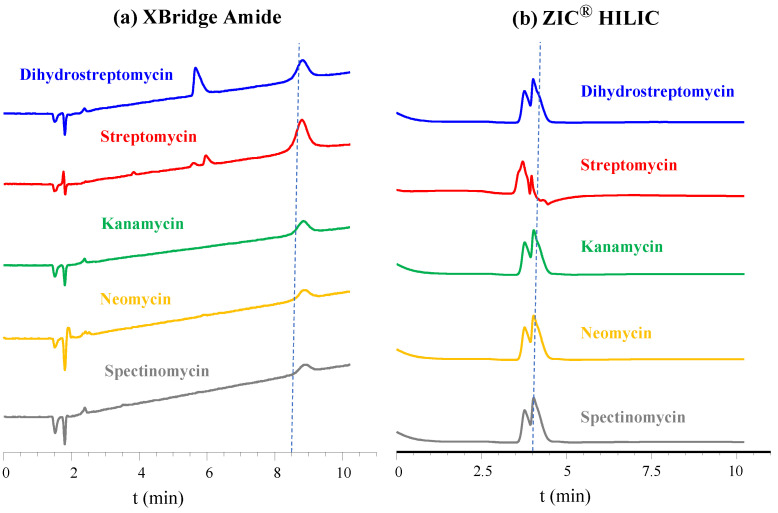
Comparison of the chromatograms obtained for each aminoglycoside with two hydrophilic stationary phases: (**a**) XBridge Amide and (**b**) ZIC^®^-HILIC. Mobile phase consisted of a mixture of 100 mM aqueous ammonium formate and acetonitrile.

**Figure 2 molecules-29-03210-f002:**
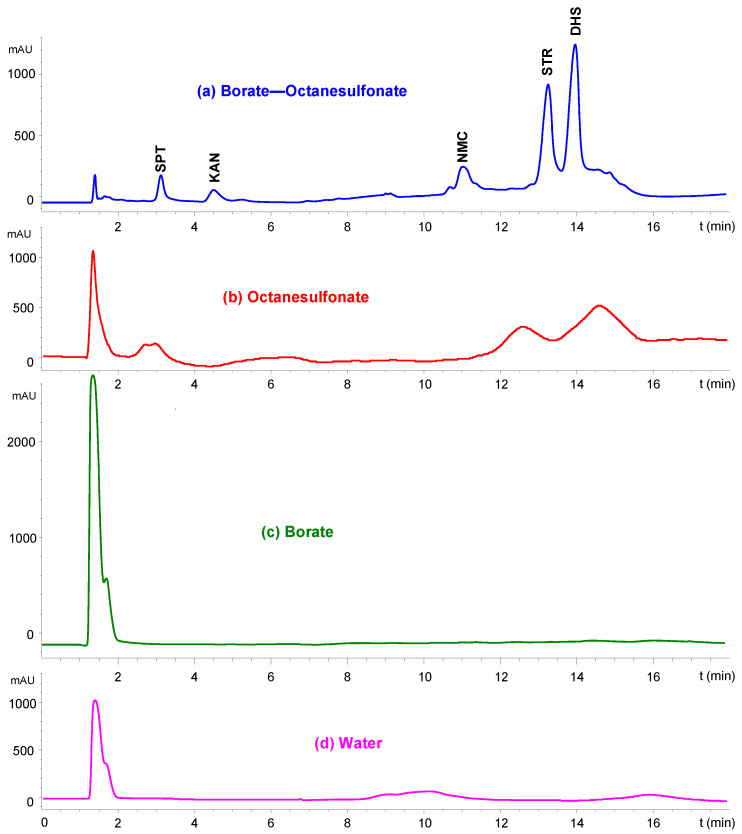
IP-LC-UV chromatograms obtained from mobile phases containing borate and octanesulfonate (**a**), octanesulfonate (**b**), borate (**c**) and neither borate nor octanesulfonate (**d**).

**Figure 3 molecules-29-03210-f003:**
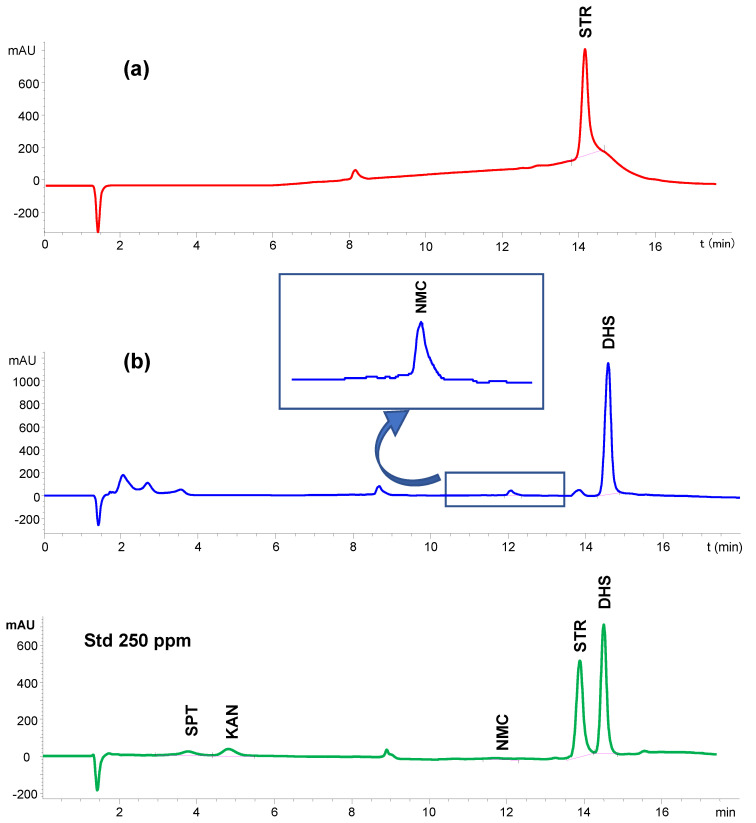
IP-LC-UV chromatograms corresponding to the analysis of two commercially available formulations: (**a**) powder for injection containing 1 g of streptomycin sulphate and (**b**) tablets containing 21 mg of Dihydrostreptomycin sulphate and 39 mg of Neomycin sulphate per tablet.

**Figure 4 molecules-29-03210-f004:**
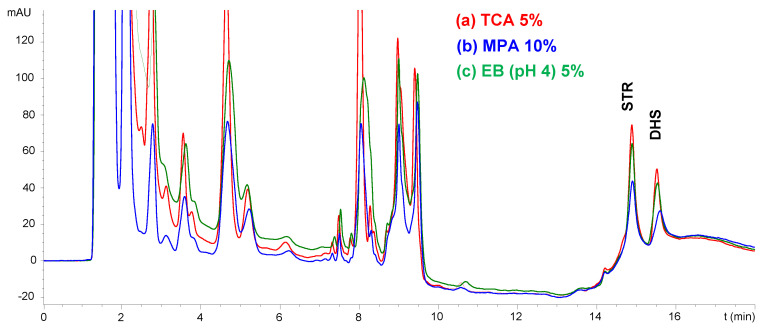
IP-LC-UV chromatograms obtained after precipitation of proteins of a serum sample spiked with STR and DHS at 25 mg L^−1^ treated with the same volume of a solution of TCA 5% (red line), MPA 10% (blue line) and an extraction buffer pH 4 (green line).

**Figure 5 molecules-29-03210-f005:**
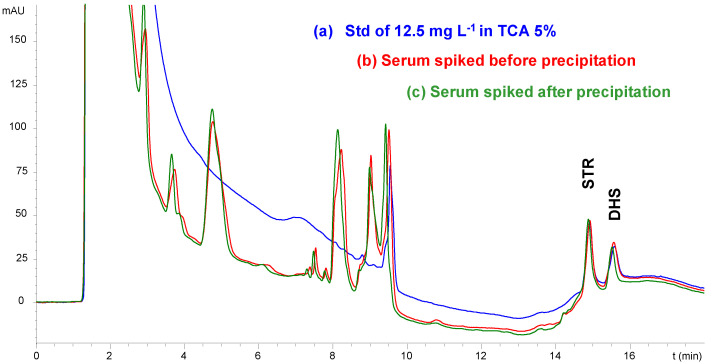
IP-LC-UV chromatograms corresponding to (a) standard of STR and DHS at concentration of 12.5 mg L^−1^ in TCA (blue line), (b) in serum sample spiked before precipitation (serum/TCA 2:1) (red line) and (c) in serum sample spiked after precipitation (serum/TCA 2:1) (green line).

**Figure 6 molecules-29-03210-f006:**
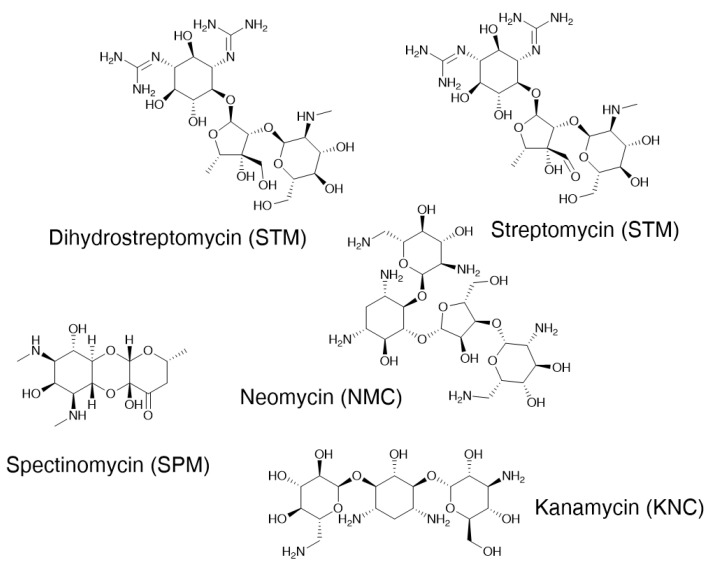
Structures of the aminoglycosides included in this study.

**Table 1 molecules-29-03210-t001:** Analytical characteristics of the proposed methodology. Repeatability and reproducibility as RSD at two different concentration levels (LOQ and 1000 mg L^−1^).

	SPM	KNC	NMC	STM	DHS
**Slope** **±** **SD**	2.6 ± 0.1	2.1 ± 0.1	6.4 ± 0.5	12.0 ± 0.8	21.1 ± 0.3
**Intercept SD**	36 ± 42	11 ± 58	43 ± 61	28 ± 42	40 ± 63
**r^2^**	0.999	0.996	0.991	0.992	0.999
**LOD (mg L^−1^)**	40	37	64	6	3
**LOQ (mg L^−1^)**	128	107	192	21	14
**Repeatability (%)** **(LOQ)**	5.0	8.6	9.9	8.0	1.5
**Repeatability (%)** **(1000 mg L^−1^)**	3.5	2.7	2.3	1.2	0.9
**Reproducibility (%)** **(LOQ)**	14.6	12.1	9.8	17.2	10.2
**Reproducibility (%)** **(1000 mg L^−1^)**	5.9	3.5	1.8	6.8	1.9

**Table 2 molecules-29-03210-t002:** Analytical results of the analysis of aminoglycosides in two different pharmaceutical formulations.

Pharmaceutical Formulation	Analyte	Labelled	Added	Found	Recovered (%)
Streptomycine (sulphate) Generic	STR	1 g	0	0.982 ± 0.005	98
			0.100 g	1.103 ± 0.005	100
			0.250 g	1.260 ± 0.006	101
Sulfintestin Neomycine^®^	DHS	39 mg/tablet	0	41.5 ± 0.2	106
			3.9 mg	42.6 ± 0.2	99
			9.7 mg	48.9 ± 0.2	100
	NMC	21 mg/tablet	0	19.5 ± 0.1	93
			2.1 mg	22.8 ± 0.2	98
			5.6 mg	26.0 ± 0.2	99

## Data Availability

The original contributions presented in the study are included in the article; further inquiries can be directed to the corresponding author.

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
