# Peer review of "Determination of Aminoglycosides by Ion-Pair Liquid Chromatography with UV Detection: Application to Pharmaceutical Formulations and Human Serum Samples"

_molecules, 2024, doi:10.3390/molecules29133210_

Round 1

Reviewer 1 Report

Comments and Suggestions for Authors

The authors developed an ion pair HPLC-UV method for analysis of five aminoglycoside antibiotics in pharmaceutical formulations and human serum samples (Sample treatment by protein precipitation).

 In general, the manuscript involved a good piece of work, showing new results and I recommend its acceptance pending language and grammatical improvement. Also, double check the reference format to match with the journal guidelines. One more thing, in Fig. 1a, isn’t possible to correct the baseline? It is possible using Origin software; however, I am not sure if you have a software that could help improve the baseline of LC chromatograms.

Comments on the Quality of English Language

It is recommended to improve the language quality via native english speaker or professional editing service.

Author Response

The authors developed an ion pair HPLC-UV method for analysis of five aminoglycoside antibiotics in pharmaceutical formulations and human serum samples (Sample treatment by protein precipitation).

In general, the manuscript involved a good piece of work, showing new results and I recommend its acceptance pending language and grammatical improvement.

We thank the reviewer for its kind comments

Also, double check the reference format to match with the journal guidelines.

References have been formatted as established by the journal guidelines

One more thing, in Fig. 1a, isn’t possible to correct the baseline? It is possible using Origin software; however, I am not sure if you have a software that could help improve the baseline of LC chromatograms.

Thanks for your comments. Unfortunately, we don´t have Origin or any other similar software to correct the baseline.

Comments on the Quality of English Language

It is recommended to improve the language quality via native english speaker or professional editing service.

The revised manuscript has been intensively checked and edited by a native English speaker

Reviewer 2 Report

Comments and Suggestions for Authors

The authors focused on an interesting topic, namely the separation and determination of selected aminoglycosides. However, in its present form, the paper does not have much scientific value and is unsuitable for publishing. A few major revisions should improve the paper's quality.

Major revisions:

1) Paragraph 2.1: What conditions were used during the HILIC analysis? Does it was an isocratic or gradient elution program? What was the percentage of organic solvent? Did the authors try to change those conditions to obtain better separation?

2) Figure 5: Chromatogram b) and c) shows that there are no visible differences between obtained chromatograms before and after protein precipitation. So, why sample preparation step is necessary since do not remove any interferences? 

3) The authors often use phase “the optimized IP-LC-UV procedure” but do not show the process of conditions selection not even more so numerical conditions optimization. The research should be filled with data about the influence of different chromatographic conditions on the separation and limit of detection of the analyzed compounds.

Minor revisions:

4) The Section 2 title should be changed to “Results and Discussion”.

5) Line 128-129: Wrong figure references.

6) Figure 1: The spacing between chromatograms is too big and it makes it difficult to compare retention times.

7) Figure 6: The presented structures are of low quality. The authors should improve it.

Comments on the Quality of English Language

The paper is full of minor language and terminological mistakes.

Author Response

The authors focused on an interesting topic, namely the separation and determination of selected aminoglycosides. However, in its present form, the paper does not have much scientific value and is unsuitable for publishing. A few major revisions should improve the paper's quality.

Major revisions:

1) Paragraph 2.1: What conditions were used during the HILIC analysis? Does it was an isocratic or gradient elution program? What was the percentage of organic solvent? Did the authors try to change those conditions to obtain better separation?

Of course, we tried different aqueous mobile phases and different initial conditions including different isocratic and gradient conditions. However, changes in the separation of the compounds were not observed. For this reason, specific conditions were not mentioned as they all result in similar results. This fact has been clarified in the revised manuscript.

2) Figure 5: Chromatogram b) and c) shows that there are no visible differences between obtained chromatograms before and after protein precipitation. So, why sample preparation step is necessary since do not remove any interferences?

In both chromatograms precipitation of proteins was carried out. The difference between chromatogram b) and c) is that samples were spiked before and after protein precipitation. This fact has been clarified in the Figure 5 legend.

3) The authors often use phase “the optimized IP-LC-UV procedure” but do not show the process of conditions selection not even more so numerical conditions optimization. The research should be filled with data about the influence of different chromatographic conditions on the separation and limit of detection of the analyzed compounds.

Optimization of the separation is described in section 2.2. Different ionic pair compounds were studied, and composition and variation of the mobile phases was also studied. A paragraph has been added clarifying what and how was optimized.

Minor revisions:

4) The Section 2 title should be changed to “Results and Discussion”.

Done.

5) Line 128-129: Wrong figure references.

Figure references have been changed in the revised manuscript.

6) Figure 1: The spacing between chromatograms is too big and it makes it difficult to compare retention times.

Figure has been modified to make easier to compare retention times.

7) Figure 6: The presented structures are of low quality. The authors should improve it.

            Figure 6 has been remade from scratch in the revised manuscript

Comments on the Quality of English Language

The paper is full of minor language and terminological mistakes.

The revised manuscript has been intensively checked and edited by a native English speaker

Reviewer 3 Report

Comments and Suggestions for Authors

 Dear Authors,

 I have reviewed your Manuscript titled: ”Determination of aminoglycosides by ion pair liquid chromatography with UV detection: application to pharmaceutical formulations and human serum samples.” and concluded that it is an interesting study dealing with the topic that is relevant for the readership of the journal Molecules. However, some issues need to be addressed.

1.     I suggest changing the title to ”Determination of aminoglycosides by ion-pair liquid chromatography with UV detection: application to pharmaceutical formulations and human serum samples”, the dot at the end should be deleted.

2.     In the introduction, a few sentences about the comparison of liquid chromatography and UV spectrophotometrics should be noted, as well as advances in chromatography techniques and cite references https://doi.org/10.1080/14786410801886682  https://doi.org/10.1016/j.jics.2024.101173 and other appropriate.

3.     In Figure 2d, can you explain the peak appearing at 1.8 min when water is recorded?

4.     In Table 1, please correct the second-row Intercept ± SD, and Repeatability (%) (1000 mg L-1), the bracket should be closed.

5.     In Table 1, please check if it is R2 or r?

6.     In all Figures please add t (min) on the x-axis and 0 at the beginning of the x-axis.

7.     The quality of Figure 6 should be improved.

Author Response

I have reviewed your Manuscript titled: ”Determination of aminoglycosides by ion pair liquid chromatography with UV detection: application to pharmaceutical formulations and human serum samples.” and concluded that it is an interesting study dealing with the topic that is relevant for the readership of the journal Molecules. However, some issues need to be addressed.

We thank the reviewer for its kind comments

  1. I suggest changing the title to ”Determination of aminoglycosides by ion-pair liquid chromatography with UV detection: application to pharmaceutical formulations and human serum samples”, the dot at the end should be deleted.

            The title has been changed in the revised manuscript as suggested by the reviewer

  1. In the introduction, a few sentences about the comparison of liquid chromatography and UV spectrophotometrics should be noted, as well as advances in chromatography techniques and cite references https://doi.org/10.1080/14786410801886682 https://doi.org/10.1016/j.jics.2024.101173 and other appropriate.

A new paragraph has been added to the introduction of the revised manuscript as asked by the reviewer. We have also cited the above-mentioned articles.

  1. In Figure 2d, can you explain the peak appearing at 1.8 min when water is recorded?

The peak that appears at 1.8 min when water is used as mobile phase correspond to unretained target compounds. It should be noted that the signal is higher when borate is used as is showed in fig 2.c. Separation is only achieved when octanesulfonate is added to mobile phase (fig 2.a and 2.b). This fact has been clarified in the revised manuscript

  1. In Table 1, please correct the second-row Intercept ± SD, and Repeatability (%) (1000 mg L-1), the bracket should be closed.

Done

  1. In Table 1, please check if it is R2or r?

            Done

  1. In all Figures please add t (min) on the x-axis and 0 at the beginning of the x-axis.

            Done

  1. The quality of Figure 6 should be improved.

Figure 6 has been remade from scratch in the revised manuscript

Reviewer 4 Report

Comments and Suggestions for Authors

The background of the work is sufficient to raise the need for the development of an analytical method, within an application problem.

The recommended methodology complies with standards of good laboratory practices, in terms of its methodology and development.

The validation of the analytical method presents some deviations from the specifications recommended by international guides, in the precision parameters (repeatability and reproducibility), as well as in accuracy. This is important, since although the authors propose that the method can be applied to evaluate pharmaceutical forms and biological samples, the results of the mentioned parameters are useful in biological methods, but are not sufficient for quantification in pharmaceutical forms, since The variability is higher than those accepted by international validation guides, which recommend precision up to 15 or 20%, and accuracy up to 20 or 25% in bioanalytical methods, however, for physicochemical methods (this is how it would be classified for analysis of pharmaceutical samples), the precision should have a CV% less than 5% the same as the accuracy or even lower.

The method can be applied to various types of samples, but has limitations for pharmaceutical samples.

The graphs presented are adequate, as is the validation summary shown in tables.

The work can be published after making some minor revisions.

Author Response

The background of the work is sufficient to raise the need for the development of an analytical method, within an application problem.

The recommended methodology complies with standards of good laboratory practices, in terms of its methodology and development.

We thank the reviewer for its kind comments

The validation of the analytical method presents some deviations from the specifications recommended by international guides, in the precision parameters (repeatability and reproducibility), as well as in accuracy. This is important, since although the authors propose that the method can be applied to evaluate pharmaceutical forms and biological samples, the results of the mentioned parameters are useful in biological methods, but are not sufficient for quantification in pharmaceutical forms, since The variability is higher than those accepted by international validation guides, which recommend precision up to 15 or 20%, and accuracy up to 20 or 25% in bioanalytical methods, however, for physicochemical methods (this is how it would be classified for analysis of pharmaceutical samples), the precision should have a CV% less than 5% the same as the accuracy or even lower.

The method can be applied to various types of samples but has limitations for pharmaceutical samples.

Accuracy of the method proposed is below 5%, as can be seen in Table 2, for spiked samples, recoveries were always ≤2%. For non-spiked samples we can’t be sure of the real amount present. In the case of repeatability and reproducibility these were below or close to 5% when the studies were carried out at a concentration level similar to that present in the pharmaceutical formulation analysed. Only, when the study was carried out at the concentration level of the LOQ, values were clearly over that value.

The graphs presented are adequate, as is the validation summary shown in tables.

The work can be published after making some minor revisions.

Round 2

Reviewer 2 Report

Comments and Suggestions for Authors

In my opinion, the authors complied with the comments. It improves the quality of the paper and I suggest that it should be accepted in its present form.